# Subset-Saturated Transition Cost Partitioning for Optimal Classical Planning

**Dominik Drexler** and **David Speck** and **Robert Mattmüller**

University of Freiburg, Germany

{drexlerd, speckd, mattmuel}@informatik.uni-freiburg.de

## Abstract

Cost partitioning admissibly combines the information from multiple heuristics for state-space search. We use a greedy method called saturated cost partitioning that considers the heuristics in sequence and assigns the minimal fraction of the remaining costs that it needs to preserve the heuristic estimates. In this work, we address the problem of using more expressive transition cost functions with saturated cost partitioning to obtain stronger heuristics. Our contribution is subset-saturated transition cost partitioning that combines the concepts of using transition cost functions and prioritizing states that look more important during the search. Our empirical evaluation shows that this approach still causes too much computational overhead but leads to more informed heuristics.

## Introduction

Heuristic search with a lower bounding heuristic is one of the most promising techniques to solve challenging planning problems optimally. We consider cost partitioned heuristics that combine information from multiple sources. Cost partitioned heuristics use the theory of cost partitioning (Katz and Domshlak 2008). A cost partitioning distributes each transition's cost over multiple heuristics such that the sum does not exceed the original cost. If each heuristic is admissible for the assigned transition costs, then the sum of heuristic estimates is admissible. The most effective approach to compute cost partitioned heuristics is saturated cost partitioning (Seipp, Keller, and Helmert 2017). Saturated cost partitioning considers the heuristics in sequence and assigns the minimum fraction of the remaining costs that it needs to preserve the heuristic estimates. Recently, saturated cost partitioning with restrictive operator cost functions yielded state-of-the-art performance (Seipp and Helmert 2019).

In this work, we address the problem of using more expressive transition cost functions with saturated cost partitioning to obtain stronger heuristics. Transition cost functions require computationally more demanding representations compared to operator cost functions but allow computing more informed cost partitioned heuristics. The baseline approach is saturated transition cost partitioning by Keller et al. (2016). It uses the minimal fraction of the remaining cost required to preserve the heuristic estimates, which often results in an increased computational effort but also in more informed heuristics.

We can decrease the computational effort by considering a larger solution set. Moving from operator to transition cost functions typically already increases the solution set because every operator cost function is a transition cost function (but not vice versa). Another way of increasing the size of the solution set is by preserving the heuristic estimates of a subset of states. The concepts of preserving the heuristic estimates of a subset of states were introduced with subset-saturated (operator) cost partitioning on restrictive operator cost functions (Seipp and Helmert 2019). Preserving the heuristic estimates of a subset of states has shown to yield more accurate cost partitioned heuristics.

Our contribution is subset-saturated transition cost partitioning that combines saturated transition cost partitioning with the concepts of preserving the heuristic estimates of a subset of states. Even though subset-saturated transition cost partitioning does not necessarily compute better cost partitioned heuristics, we provide an empirical analysis to show that this is often the case in practice. We derive a mechanism for selecting transition cost functions from the solution set that trades heuristic accuracy with performance and follows the principle of prioritizing a subset of states.

## Background

In this chapter we define, describe and discuss the concepts and ideas that form the foundation for subset-saturated transition cost partitioning.

### Planning Tasks

We consider the $SAS^+$ planning formalism (Bäckström and Nebel 1995). A $SAS^+$ planning task is a 5-tuple that consists of a set of finite domain variables that induce a set of states, a finite set of operators (or actions) that induce a finite set of transitions, an initial state, a goal condition and a function that describes the cost of applying each operator. Each planning task compactly encodes a transition system with weights assigned to transitions.

## Transition Systems

A transition system describes the dynamics of a state-based system. Transition systems (Seipp and Helmert 2019) are also called state spaces.

**Definition 1** (Transition System). *A Transition System $\mathcal{T}$ is a directed, labeled graph defined by a finite set of states $S(\mathcal{T})$, a finite set of labels $L(\mathcal{T})$, a finite set $T(\mathcal{T})$ of labeled transitions $s \xrightarrow{l} s$ with $s, s' \in S(\mathcal{T})$ and $l \in L(\mathcal{T})$, an initial state $s_0(\mathcal{T}) \in S(\mathcal{T})$ and a set $S_\star(\mathcal{T}) \subseteq S(\mathcal{T})$ of goal states.*

The objective in state space search is to find paths from the initial state to a goal state.

**Definition 2** (Paths and Goal Paths (Seipp and Helmert 2019)). *Let $\mathcal{T}$ be a transition system. A path from $s \in S(\mathcal{T})$ to $s' \in S(\mathcal{T})$ is a sequence of transitions from $T(\mathcal{T})$ of the form $\pi = \left\langle s^0 \xrightarrow{l_1} s^1, \ldots, s^{n-1} \xrightarrow{l_n} s^n \right\rangle$, where $s^0 = s$ and $s^n = s'$. The length of $\pi$ denoted by $|\pi|$, is $n$. The empty path (of length 0) is permitted if $s = s'$. $\pi$ is called a goal path, if it is a path to any goal state $s' \in S_\star(\mathcal{T})$.*

In the context of classical planning and cost partitioning, it is convenient to allow assignment of costs to transitions that are not necessarily unit costs. Transition cost functions allow a different assignment for each transition and therefore, allows taking the application contexts of each action into account.

**Definition 3** (Transition Cost Function). *Let $\mathcal{T}$ be a transition system with transitions $T(\mathcal{T})$. A transition cost function for $\mathcal{T}$ is a function $tcf : T(\mathcal{T}) \to \mathbb{R} \cup \{-\infty, \infty\}$ that assigns costs to transitions. A transition cost function is finite if $-\infty < tcf(t) < \infty$ for all transitions $t \in T(\mathcal{T})$. It is nonnegative if $0 \le tcf(t)$ for all transitions $t \in T(\mathcal{T})$. We write $\mathcal{C}(T(\mathcal{T}))$ for the set of all transition cost functions for $\mathcal{T}$ and $\mathcal{C}_{\ge 0}(T(\mathcal{T}))$ for the set of all nonnegative transition cost functions for $\mathcal{T}$.*

A special case of a transition cost function is an operator cost function. It is a transition cost function where every transition with the same label is assigned the same cost value. Therefore, an operator cost function is a mapping from labels (or operators) to our considered codomain. The representation of an operator cost function requires worst-case space of $\Theta(|L(\mathcal{T})|)$ or $\Theta(||\Pi||)$ where $||\Pi||$ denotes the input size of a planning task. The representation of a transition cost function requires worst-case space of $\Theta(||\Pi|| \cdot 2^{poly(||\Pi||)})$.

In the context of cost partitioning, it is convenient to work with a collection of transition systems where each transition system is associated with a transition cost function. The definition of a weighted transition system captures the notion of this pairing (Seipp and Helmert 2019).

**Definition 4** (Weighted Transition System). *A weighted transition system is a tuple $\langle \mathcal{T}, tcf \rangle$ where $\mathcal{T}$ is a transition system and $tcf$ is a transition cost function for $\mathcal{T}$.*

Allowing infinities to be assigned to transitions through either operator or transition cost functions means that we must take care of arithmetic expressions that involve $+\infty$ and $-\infty$ to make the theory of cost partitioning work. Seipp

and Helmert (2019) defined two kinds of addition that make it possible to handle mixed infinities in cost partitioning.

The symbols $+$ (infix) and $\sum$ (prefix) denote the left-addition operation. Left-addition over integers is the usual addition. Expressions that contain infinities are defined as $\infty + x = \infty$ and $-\infty + x = -\infty$ for all integers $x$, including $x$ being $\infty$ or $-\infty$. This operation is associative but not commutative. Intuitively, a left addition sum that contains mixed infinities evaluates to the leftmost infinity. In cost partitioning, we use left-addition for summing up multiple heuristic values and partitioned costs (Seipp and Helmert 2019).

The symbols $\oplus$ (infix) and $\bigoplus$ (prefix) denote the path-addition operation. Path addition over integers is the usual addition. Expressions that contain infinities are defined as $x \oplus y = \infty$ iff $x = \infty$ or $y = \infty$ and $x \oplus (-\infty) = -\infty$ for all $x \ne \infty$. This operation is associative and commutative. Intuitively, a path addition sum evaluates to $+\infty$ if the path addition sum involves at least one $+\infty$. We use path-addition to define the cost of a path in a transition system (Seipp and Helmert 2019).

**Definition 5** (Cost of a path). *Let $\langle \mathcal{T}, tcf \rangle$ be a weighted transition system. The cost of a path $\pi = \left\langle s^0 \xrightarrow{l_1} s^1, \ldots, s^{n-1} \xrightarrow{l_n} s^n \right\rangle$ with $t_i = s^{i-1} \xrightarrow{l_i} s^i$ is defined as $cost(tcf, \pi) = \bigoplus_{i=1}^n tcf(t_i)$.*

Intuitively, a path of cost $-\infty$ is infinitely cheap, and a path of cost $\infty$ is non-existent.

In optimal classical planning, we are interested in finding paths with the cheapest cost to a goal. The following definition captures goal distances as functions that depend on a given state and a transition cost function. This notion follows the pairing of weighted transition systems in cost partitioning.

**Definition 6** (Goal Distances and Optimal Paths (Seipp and Helmert 2019)). *Let $\langle \mathcal{T}, tcf \rangle$ be a weighted transition system. The goal distance of a state $s \in S(\mathcal{T})$ in $\mathcal{T}$ under cost function $tcf$ is defined as $\inf_{\pi \in \Pi_\star(\mathcal{T}, s)} cost(tcf, \pi)$ where $\Pi_\star(\mathcal{T}, s)$ is the set of goal paths from $s$ in $\mathcal{T}$.*

*We write $h_\mathcal{T}^*(tcf, s)$ for the goal distance of $s$ in $\mathcal{T}$ under transition cost function $tcf$.*

*A goal path $\pi$ from $s$ in $\mathcal{T}$ is optimal under the given transition cost function $tcf$ if $cost(tcf, \pi) = h_\mathcal{T}^*(tcf, s)$.*

The empty infimum is defined as $\infty$ and follows the notion of a non-existent goal path. The infimum ensures that repeatedly taking a negative cost cycle in the transition system will evaluate to $-\infty$.

## Abstractions

Abstractions are relaxations of the behavior of a state-based system where multiple states collapse into a single abstract state (Helmert, Haslum, and Hoffmann 2007).

**Definition 7** (Abstraction). *Let $\mathcal{T}, \mathcal{T}'$ be two transition systems with the same label sets $L(\mathcal{T}) = L(\mathcal{T}')$ and let $\alpha : S(\mathcal{T}) \to S(\mathcal{T}')$, $\beta : T(\mathcal{T}) \to T(\mathcal{T}')$ be surjective functions. We say that $\mathcal{T}'$ is an abstraction of $\mathcal{T}$ with abstraction mappings $\alpha, \beta$ if (1) $\alpha(s_0(\mathcal{T})) = s_0(\mathcal{T}')$, (2) $\alpha(s_\star) \in$*

$S_\star(\mathcal{T}')$ for all $s_\star \in S_\star(\mathcal{T})$, and (3) $\alpha(s) \xrightarrow{l} \alpha(s') \in T(\mathcal{T}')$ and $\beta(s \xrightarrow{l} s') = \alpha(s) \xrightarrow{l} \alpha(s')$ for all $s \xrightarrow{l} s' \in T(\mathcal{T})$.

We refer to $\mathcal{T}$ as the concrete transition system and $\mathcal{T}'$ as the abstract transition system. We consider a special type of abstraction where every abstract state is cartesian (Seipp and Helmert 2013).

An abstraction heuristic is a function that maps each concrete state to the goal distance of its corresponding abstract state under a given transition cost function. Goal distances in the abstract transition system require an abstract transition cost function that maps every abstract transition to a value in the codomain. The abstract transition cost function maps an abstract transition to the minimal cost of every concrete transition that induces it (Keller et al. 2016).

**Definition 8** (Abstraction heuristic). *Let $\langle \mathcal{T}, tcf \rangle$ and $\langle \mathcal{T}', tcf' \rangle$ be two weighted transition systems. Let $\mathcal{T}'$ be an abstraction of $\mathcal{T}$ with abstraction mappings $\alpha, \beta$.*

*We say that $tcf'$ is the abstract transition cost function of $tcf$ that describes the cost of each abstract transition $t' \in T(\mathcal{T}')$ in the abstraction with*

$$tcf'(t') = \min\{tcf(t) \mid t \in T(\mathcal{T}) \wedge \beta(t) = t'\}$$

*The abstraction heuristic of a concrete state $s \in S(\mathcal{T})$ is the goal distance of the corresponding abstract state $\alpha(s)$ in the abstraction $\mathcal{T}'$ under the abstract transition cost function $tcf'$, i.e. $h(tcf, s) = h^*_{\mathcal{T}'}(tcf', \alpha(s))$.*

The definition of the abstract transition cost function reveals that the computation of an abstract transition cost involves a minimization over all (exponentially many) transitions of the concrete transition system that are responsible for this abstract transition. In the case of cartesian abstractions this minimization of exponentially many concrete transitions can be carried out in time that is often polynomial in the number of variables of the planning task, albeit worst-case exponential (Geißer, Keller, and Mattmüller 2016).

A heuristic is admissible if it never overestimates the goal distances in the weighted concrete transition system. Abstraction heuristics are admissible because every goal path in the concrete transition system corresponds to a goal path in the abstract transition system. The minimization in the abstract transition cost function ensures that the abstraction heuristic does not overestimate any goal distance (Keller et al. 2016). We use the $A^*$ algorithm with an admissible heuristic to find optimal goal paths (Hart, Nilsson, and Raphael 1968).

### Transition Cost Partitioning

A transition cost partitioning splits a given transition cost function into a sequence of transition cost functions such that the sum (left addition) of all transition cost functions in the sequence is upper bounded by the original transition cost function (Keller et al. 2016; Pommerening 2017).

**Definition 9** (Transition Cost Partitioning). *A transition cost partitioning for a weighted transition system $\langle \mathcal{T}, tcf \rangle$ with transition cost function $tcf \in \mathcal{C}(T(\mathcal{T}))$ is a tuple $\langle tcf_1, \ldots, tcf_n \rangle \in \mathcal{C}(T(\mathcal{T}))^n$ whose sum is bounded by $tcf$, i.e. $\sum_{i=1}^{n} tcf_i(t) \leq tcf(t)$ for all $t \in T(\mathcal{T})$.*

A transition cost partitioning induces a cost partitioned heuristic by associating each transition cost function with a heuristic and summing up the heuristic estimates of individual states. The following theorem states that the cost partitioned heuristic is admissible. Katz and Domshlak (2008) introduced cost partitioning that works on nonnegative operator cost functions. Pommerening et al. (2015) showed that general operator cost functions can be used for cost partitioning and Keller et al. (2016) defined general transition cost function for cost partitioning. Finally, Seipp and Helmert (2019) introduced left addition rules to handle mixed infinities, which led to Theorem 1 and shows that cost partitioned heuristics are admissible.

**Theorem 1.** *Admissibility Let $\mathcal{T}$ be a transition system with admissible heuristics $\langle h_1, \ldots, h_n \rangle$ and a transition cost partitioning $P(T(\mathcal{T})) = \langle tcf_1, \ldots, tcf_n \rangle \in \mathcal{C}(T(\mathcal{T}))^n$. Then $h_{P(T(\mathcal{T}))}(s) = \sum_{i=1}^{n} h_i(tcf_i, s)$ is an admissible heuristic.*

An optimal transition cost partitionings for a set of heuristics is a transition cost partitioning that provides the best heuristic estimate for a given state. The best known algorithm to compute optimal transition cost partitioning works in exponential time and is not useful in practice. In the next section, we focus on greedy algorithms based on cost saturation that consider the heuristic in sequence and builds the transition cost partitioning sequentially.

## Subset-Saturated Transition Cost Partitioning

In this section, we combine saturated transition cost partitioning (Keller et al. 2016) with subset-saturation known from subset-saturated operator cost partitioning (Seipp and Helmert 2019). We first generalize dominating cost functions (Seipp and Helmert 2019).

**Definition 10** (Dominating Transition Cost Function). *Consider two transition cost functions $tcf$ and $tcf'$ defined on the same set of transitions. We say that $tcf$ dominates $tcf'$, in symbols $tcf \leq tcf'$, if $tcf(t) \leq tcf'(t)$ for all transitions $t$.*

*We say that $tcf$ is the unique minimum of a set of transition cost functions $Cost$ if it dominates all transition cost functions in $Cost$. (Not all sets $Cost$ have a unique minimum.)*

A saturated transition cost function for a subset of states dominates a given transition cost function and preserves the heuristic estimates of a given subset of states (Seipp and Helmert 2019).

**Definition 11** (Saturated Transition Cost Function). *Consider a weighted transition system $\langle \mathcal{T}, tcf \rangle$, a set of states $S' \subseteq S(\mathcal{T})$ and a heuristic $h$ for $\mathcal{T}$. A transition cost function $stcf \in \mathcal{C}(T(\mathcal{T}))$ is saturated for $S', h$ and $tcf$ if*

*1. $stcf \leq tcf$ and*
*2. $h(stcf, s) = h(tcf, s)$, for all states $s \in S'$.*

A saturated transition cost function always exists because $tcf$ itself is a saturated transition cost function. Our definition of the saturated transition cost function allows selecting from a typically larger set of possible saturated transition

cost functions because of allowing transition cost functions instead of operator cost functions and preserving the heuristic estimates of a subset of states. We formalize this selection mechanism with a generalization of operator saturators (Seipp and Helmert 2019) to transition saturators. A transition saturator is a function that takes as an input a heuristic, the remaining transition cost function, and a subset of states and outputs a saturated transition cost function. In contrast, an operator saturator only allows operator cost functions in the input and output.

**Definition 12** (Transition Saturator). *Consider a transition system $\mathcal{T}$, a set of states $S' \subseteq S(\mathcal{T})$ and a heuristic $h$ for $\mathcal{T}$.*

*A transition saturator for $S'$ and $h$ is a partial function $saturate : \mathcal{C}(T(\mathcal{T})) \to \mathcal{C}(T(\mathcal{T}))$ such that whenever $saturate(tcf)$ is defined, it is a saturated transition cost function for $S'$, $h$ and $tcf$.*

*A transition saturator is general if its domain of definition is $\mathcal{C}(T(\mathcal{T}))$. It is nonnegative to general (NNG) if its domain of definition is $\mathcal{C}_{\geq 0}(T(\mathcal{T}))$. It is nonnegative if its domain of definition is $\mathcal{C}_{\geq 0}(T(\mathcal{T}))$ and it only produces transition cost functions in $\mathcal{C}_{\geq 0}(T(\mathcal{T}))$.*

The following definition generalizes subset-saturated operator cost partitioning by exchanging operator saturators with transition saturators. Alternatively speaking, we parameterize saturated transition cost partitioning with transition saturators and allow saturation for a subset of states.

**Definition 13** (Subset-Saturated Transition Cost Partitioning). *Consider a weighted transition system $\langle \mathcal{T}, tcf \rangle$, a set of states $S' \subseteq S(\mathcal{T})$, a nonempty sequence of heuristics $\mathcal{H} = \langle h_1, \ldots, h_n \rangle$ for $\mathcal{T}$ and a sequence $Saturate = \langle saturate_1, \ldots, saturate_n \rangle$ such that $saturate_i$ is a saturator for $S' \subseteq S(\mathcal{T})$ and $h_i$ for all $1 \leq i \leq n$.*

*The saturated transition cost partitioning $\langle tcf_1, \ldots, tcf_n \rangle$ of the transition cost function $tcf$ induced by $Saturate$ is defined as:*

$$remain_0 = tcf$$
$$tcf_i = saturate_i(remain_{i-1}) \text{ for all } 1 \leq i \leq n$$
$$remain_i = remain_{i-1} - tcf_i \text{ for all } 1 \leq i \leq n$$

The subtraction in the definition of $remain_i$ is defined in terms of left addition, i.e., $a - b := a + (-b)$ and corresponds the definition from subset-saturated operator cost partitioning.

The saturated transition cost partitioning (Definition 13) is a transition cost partitioning. The result follows from the proof that subset-saturated operator cost partitioning produces operator cost partitionings and exchanges labels with transitions (Seipp and Helmert 2019).

In general, subset-saturated cost partitioning has three major choice points that influence the accuracy of the cost partitioned heuristic. These are the set of heuristics, the order of the heuristics, and the transition saturators. In the remaining part of this section, we define generalizations of the four operator saturators and an additional transition saturator that allows avoiding computations of abstract transition weights (Definition 8).

When comparing transition saturators, we use the same concept of domination between saturated cost functions. A dominating saturated cost function is "more economical" than the cost function that it dominates because it achieves the same objective of preserving heuristic estimates but leaves a larger fraction of the remaining costs for later heuristics in sequence. This often gives better heuristic estimates but is not guaranteed due to the greediness of saturated cost partitioning (Seipp and Helmert 2019).

We generalize the comparison results of operator saturators to allow for comparison of transition saturators. We provide an additional result for comparing operator saturators with transition saturators.

**Theorem 2** (Domination by Subsets). *For a given transition system $\mathcal{T}$, heuristic $h$ for $\mathcal{T}$ and transition cost function $tcf \in \mathcal{C}(T(\mathcal{T}))$, let $STCF(X)$ be the set of saturated transition cost functions for the set of states $X \subseteq S(\mathcal{T})$, $h$ and $tcf$.*

*Let $S'' \subseteq S' \subseteq S(\mathcal{T})$. Then:*

1. *For all transition cost functions $tcf' \in STCF(S')$, there exists a transition cost function $tcf'' \in STCF(S'')$ that dominates $tcf'$.*

2. *If a transition cost function $tcf'' \in STCF(S'')$ is the unique minimum of $STCF(S'')$, then $tcf''$ dominates all transition cost functions in $STCF(S')$.*

**Proof:** Statement 1 follows from Definition 11 where we allow assigning lower saturated costs to transitions that start at or end at states outside the subset if it does not conflict with preserving the heuristic estimates of states within the subset. Such transitions may exist exclusively in $S''$, because $S'' \subseteq S'$. Statement 2 follows from Definition 10 of a unique minimum and the first statement. ∎

The theorem is an analog extension of the theorem about domination by subsets on operator saturators (Seipp and Helmert 2019). It shows that transition saturators for $S'' \subseteq S'$ are more economical than transition saturators for $S'$. Since sets of saturated transition cost functions do not necessarily have a unique minimum, it is not guaranteed that a minimal saturated transition cost function for $STCF(S'')$ dominates all saturated transition cost functions in $STCF(S')$. This stronger notion of dominance requires that $STCF(S'')$ has a unique minimum and is part 2 of the theorem.

The second way to obtain more economical transition saturators uses saturator composition where the output of a saturator is applied to the input of another saturator (Seipp and Helmert 2019).

**Theorem 3** (Domination by Composition). *Let $saturate_1$ and $saturate_2$ be transition saturators for the same transition system $\mathcal{T}$, state set $S' \subseteq S(\mathcal{T})$ and heuristic $h$. Let $saturate_{12} : \mathcal{C}(T(\mathcal{T})) \to \mathcal{C}(T(\mathcal{T}))$ be the composition of these saturators, i.e. $saturate_{12}(tcf) = saturate_2(saturate_1(tcf))$ for all transition cost functions $tcf \in \mathcal{C}(T(\mathcal{T}))$.*

*Then $saturate_{12}$ is a transition saturator for $\mathcal{T}$, $S'$ and $h$, and for all transition cost functions $tcf \in \mathcal{C}(T(\mathcal{T}))$, $saturate_{12}(tcf)$ dominates $saturate_1(tcf)$.*

**Proof:** Follows directly from Definition 12, where we require the output of a transition saturator to dominate its input. ∎

The composition is the reason why we consider saturators in the general case. The inner saturator can output negative costs that the outer saturator has to handle. The composition with $saturate_2(\max(0, saturate_1(tcf)))$ ensures that $saturate_2$ is considered in the NNG case. This does not violate Definition 11 statement 1 because $tcf$ is the remaining cost function and nonnegative. The following theorem states that allowing saturators to output transition cost functions makes them more economical.

**Theorem 4** (Domination by Expressiveness). *Let $saturate_o$ be an operator saturator for transition system $\mathcal{T}$, state set $S' \subseteq S(\mathcal{T})$ and heuristic $h$. Then there exists a transition saturator $saturate_t$ for $\mathcal{T}$, $S'$, and $h$ such that $saturate_t(ocf)$ dominates $saturate_o(ocf)$ for all operator cost functions $ocf \in \mathcal{C}(L(\mathcal{T}))$.*

**Proof:** The output of any operator saturator is an operator cost function and a special case of the output of a transition saturator. Hence, there exists a transition saturator that returns the same saturated transition cost function as the operator saturator. ∎

In other words, for each operator saturator, we can construct a transition saturator that produces a dominating saturated transition cost function. In the rest of this chapter, we define such a transition saturator for each known operator saturator. We introduce a new transition saturator that is explicitly used for transition cost functions and improves the performance of computing heuristic estimates. Whenever we provide additional information about the experimental setup, we describe the setup that allows for a fair comparison with operator saturators.

### General transition saturators

In the definition of each transition saturator, we consider a weighted concrete transition system $\langle \mathcal{T}, tcf \rangle$ where $tcf$ describes the current remaining transition cost function and an abstraction heuristic $h$ for $\langle \mathcal{T}, tcf \rangle$ with underlying weighted abstract transition system $\langle \mathcal{T}', tcf' \rangle$.

**Saturate for all states (all$_t$)** The $all_t$ transition saturator preserves the heuristic estimates of all states and is the one that was considered previously by Keller et al. (2016). It computes the unique minimum saturated transition cost function $mstcf$ by setting the consistency constraint $h(tcf, s) \leq h(tcf, s') + mstcf(t)$ tight for all transitions $t = s \xrightarrow{l} s' \in T(\mathcal{T})$. This can be enforced by setting

$$mstcf(t) = h(tcf, s) \ominus h(tcf, s')$$

Seipp, Keller, and Helmert (2020) defined the operator $\ominus$ in the context of computing the minimum saturated operator cost function $msocf$. The operator $\ominus$ behaves like the regular subtraction in the finite case and handles infinities as $x \ominus y = -\infty$ iff $x = -\infty$ or $y = \infty$ and $x \ominus y = \infty$ iff $x = \infty \neq y$ or $x \neq -\infty = y$. The minimum saturated

operator cost function $msocf$ sets the consistency constraint tight for at least one transition of each operator, i.e.,

$$msocf(l) = \sup_{s \xrightarrow{l} s' \in T(\mathcal{T})} h(tcf, s) \ominus h(tcf, s')$$

where the empty supremum is defined as $-\infty$ and $tcf$ is an operator cost function in saturated operator cost partitioning. Seipp, Keller, and Helmert (2020) show that the operator $\ominus$ computes the minimal saturated cost for each transition and the supremum generalizes over all transitions with the same label such that context information does not need to be tracked. Hence, the $mstcf$ computes the unique minimum among all saturated transition cost functions that preserve the heuristic estimates of all states.

**Saturate for all states (spd$_t$)** The transition saturator $spd_t$ is nonnegative and preserves the heuristic estimates of all states. Its name is derived from the Shortest Path Discovery (SPD) problem (Szepesvári 2004), where we are given a transition system, a function $query$ that returns the cost of a transition, and a lower bound on the true transition weights. The objective is to find the exact goal distance of each state[1], such that the number of evaluations of the function $query$ is as small as possible.

The SPD problem occurs in the saturated transition cost partitioning algorithm as follows: For the next abstraction heuristic in sequence, we do not know the abstract transition weights. But we can compute their weights using Definition 8. Our experiments have shown that computing all abstract transition weights is a performance bottleneck. According to Definition 13, we know that $remain_i$ is nonnegative if $remain_0$ is nonnegative (as in classical planning). Therefore, a lower bound for each abstract transition weight is zero[2].

A nonnegative lower bound is important because it allows using Dijkstra's algorithm for goal distance analysis. The lower bound can avoid the computation of the exact abstract transition weight if it does not shorten goal distances during goal distance analysis. Consider the case that Dijkstra's expands an abstract transition. If the lower bound on the abstract transition weight decreases the currently known goal distance of the source state, then we have to compute the exact transition weight. Otherwise, we keep the lower bound and proceed with the next abstract transition. The lower bound in the else case does not introduce shortcuts because: If the goal path that uses the transition with the lower bound on the cost is not the current cheapest path for the source state, then the same path that uses the exact transition weight is also not the current cheapest path for the source state.

**Saturate for reachable states (reach$_t$)** The transition saturator $reach_t$ is a generalization of the operator saturator

---

[1] s-t-path in the original problem definition

[2] It is possible to extract a more accurate lower bound, i.e., an operator cost function of the remaining transition cost function in $\mathcal{O}(|L(\mathcal{T})|)$ time when using decision diagrams.

*reach$_o$* (Seipp and Helmert 2019). It preserves the heuristic estimates of all states that are reachable in a forward search. The concrete preimage $S'$ of all states that are reachable in the abstract transition system overapproximates the set of reachable states. The set of unreachable states is $S(\mathcal{T}) \setminus S'$. We set the heuristic estimate of each state $s \in S(\mathcal{T}) \setminus S'$ to $h(tcf, s) = -\infty$ because they are never visited in a forward search (Seipp and Helmert 2019). These modified heuristic estimates will remain for all subsequent saturators in a composition to exclude unreachable states from the subset of states. Finally, apply *all$_t$* on the modified heuristic estimates to obtain the unique minimum saturated transition cost function for $S'$.

**Saturate for a perimeter (perim$_t$)**   The transition saturator *perim$_t$* is a generalization of the operator saturator *perim$_o$* (Seipp and Helmert 2019) and preserves the heuristic estimates of all states that are within a perimeter of $k$ to a goal. The idea is that it is more important to preserve heuristic estimates of states that are closer to a goal (e.g., Holte et al., 2004; Torralba, Linares López, and Borrajo, 2018). The set of states within perimeter $k > 0$ is $S_k = \{s \in S(\mathcal{T}) \mid h(tcf, s) \leq k\}$. In the abstract transition system $\mathcal{T}'$ the set of states $S_k$ corresponds to all abstract states $s \in \mathcal{T}'$ with $h^*_{\mathcal{T}'}(tcf', s) \leq k$.

To efficiently compute a saturated transition cost function, we only allow for nonnegative transition cost functions in the input and cap the heuristic estimates at $k = h(tcf, s_{\mathcal{I}})$ where $s_{\mathcal{I}}$ is the initial state (Seipp and Helmert 2019). Finally, apply *all$_t$* on the capped heuristic estimates to obtain a saturated transition cost function for $S_k$.

**Saturate for a single state (lp$_t$)**   The transition saturator $lp_t$ is a generalization of the operator saturator $lp_o$ (Seipp and Helmert 2019) and preserves the heuristic estimate of only a single state $s \in S(\mathcal{T})$. The set of possible saturated transition cost functions to pick from does not have a unique minimum. We use an adapted version of the linear programming formulation of the $lp_o$ operator saturator to choose from the set of possible saturated transition cost functions. The LP uses the same LP-trick to encode the heuristic estimates from the saturated transition cost function.

$$H_a \leq 0 \qquad \text{for all } a \in S_\star(\mathcal{T}') \tag{1}$$

$$H_a \leq C_t + H_b \text{ for all } a \xrightarrow{l} b = t \in T(\mathcal{T}') \tag{2}$$

$$C_t \leq tcf'(t) \quad \text{for all } t \in T(\mathcal{T}') \tag{3}$$

$$C_t \leq C_l \qquad \text{for all } a \xrightarrow{l} b = t \in T(\mathcal{T}') \tag{4}$$

$$H_{\alpha(s)} = h(s) \tag{5}$$

The variables $C_t$ encode the saturated transition cost function, the variables $C_l$ encode the saturated operator cost function, and the variables $H_a$ encode the heuristic estimate $h^*_{\mathcal{T}'}$. We also choose the objective of minimizing the sum of used operator costs $\min \sum_{l \in L(\mathcal{T})} C_l$ where we only include labels with finite saturated operator costs. Our LP formulation differs from the LP formulation of the operator saturator $lp_o$ by allowing transition cost function in the input and the output.

The objective function requires a preprocessing, where we compute the saturated transition cost of all transitions that result in either $\infty$ or $-\infty$. We first set the heuristic estimates of every state in the preimage of each abstract state that is unreachable from the abstract state $\alpha(s)$ to $-\infty$ (similar idea as described in *reach$_t$*). Then, we apply $mstcf$ to all transitions $t$ that evaluate to $mstcf(t) \in \{-\infty, \infty\}$. Finally, we restrict the LP to contain only abstract states with finite heuristic estimates together with their incident abstract transitions. If there is an abstract transition with label $l$ and a constraint of type 2, then this label $l$ is part of the objective function. After solving the LP, we extract the saturated cost of every transition from the variable $C_t$ of the corresponding abstract transition.

In our experiments, we preserve the heuristic estimate of the initial state.

### Nonnegative transition saturators

We obtain a nonnegative transition saturator by applying $stcf(t) = \max(0, stcf(t))$ for each transition $t \in T(\mathcal{T})$ on the output $stcf$ of the NNG transition saturator. We denote the nonnegative transition saturator with an additional superscript, e.g., *perim$_t^+$* for the nonnegative transition saturator of the transition saturator *perim$_t$*. It is possible to obtain similar theoretical comparison results on nonnegative transition saturators.

### Selecting runtime efficient saturator outputs

A problem with the current definitions of our transition saturators is that their output does often not perform well in our experiments. The runtime of the subtraction in Definition 13 and the size of the representation of $remain_i$ depends on the saturated transition cost function. Therefore, selecting a saturated transition cost function that performs better in the tradeoff between leaving more remaining costs for subsequent heuristics and lowering the computational overhead is necessary.

The first solution idea is to generalize saturated transition costs over operators as described in the minimum saturated operator cost function (section *all$_t$*). If we do this for all operators, then we obtain subset-saturated operator cost partitioning. Operator costs can be subtracted efficiently from a transition cost function when using decision diagrams.

The second solution idea is to decrease the number of transitions for which the saturator output $stcf(t) \neq 0$ because subtraction of zero is trivial. We can achieve this by considering the nonnegative case or considering other subsets of states. Purely considering the nonnegative case is not an option because negative costs have shown to make the heuristics stronger (Pommerening et al. 2015).

In our experiments, we use the following mechanism to select a saturated transition cost function: If an operator $o$ holds that the saturated transition cost function assigns the same value to every transition with label $o$, then we use the first solution idea because the same remaining cost is available for subsequent heuristics. Otherwise, we use a weaker notion of the second solution idea where we replace each

$stcf(t) = -\infty$ by 0. We found it prohibitively expensive in our experiments to saturate with $stcf(t) = -\infty$. The saturated transition cost $stcf(t) = -\infty$ typically occurs in dead end states or unreachable states. Intuitively speaking, this replacement corresponds to focus the computational effort on the subset of states $S'$. It preserves the same heuristic estimates but leaves fewer remaining costs for subsequent heuristics. We denote transition saturators that use this selection mechanism with additional superscript $r$, e.g. $reach_t^r$.

The proposed solution ideas enable many options to design selection mechanisms, which we leave open for future work.

### Negative costs in saturator inputs

We conclude the section of transition saturators with a discussion of the concern about the heuristic reevaluation under different types of transition cost functions. Transition saturators also require the ability to reevaluate the heuristic under the given input transition cost function. The first reevaluation under $remain_i$ is generally not too costly because $remain_i$ is always nonnegative if the cost function given with the planning task is nonnegative (as in classical planning). However, if a transition saturator saturates for a subset of states $S'$ and we later reevaluate the heuristic for states $s \notin S'$, then this requires algorithms like Bellman-Ford if the saturated transition cost function contains negative costs (Seipp and Helmert 2019).

In contrast to the operator saturators, a transition saturator has the ability to tighten the consistency constraint set to each transition. In this case, a heuristic does not change its estimates after reevaluation under the saturated transition cost function. Therefore, we can directly extract the heuristic from the output of the saturator. However, when selecting a saturated transition cost function that does not tighten each consistency constraint, then the heuristic might change after reevaluation. Seipp and Helmert (2019) found it prohibitively expensive to reevaluate the heuristic with Bellman-Ford and they suggest directly extracting a heuristic lower bound from the output of the saturator, trading heuristic accuracy with performance.

## Experiments

We implemented all transition saturators into the Fast Downward planning system (Helmert 2006). Similar to modern symbolic search planners, we utilize decision diagrams for compact representation and computation of sets of states (Kissmann and Edelkamp 2014; Torralba et al. 2014; Speck, Geißer, and Mattmüller 2018). More specifically, we use the CUDD library (Somenzi 1994) for Binary Decision Diagrams with the default fast-downward variable order to represent and compute transition cost functions. We conducted experiments with the Downward Lab toolkit (Seipp et al. 2017) on Intel Xeon E5-2650v2 2.60GHz processors with 64GB DDR3 1866MHz ECC registered memory. The benchmark set consists of all 1827 instances from the optimization track of the 1998-2018 International Planning Competitions that do not have conditional effects. Each task was limited to a single core with 4GB of memory and a time limit of 30 minutes. We consider the same set of abstraction as the initial work on cost saturation with operator saturators (Seipp and Helmert 2019) that consists of all CEGAR abstractions with the goal and landmark diversification technique (Seipp and Helmert 2014), the systematic pattern databases (Pommerening, Röger, and Helmert 2013) and the pattern databases found by hill-climbing (Haslum et al. 2007). We order the abstractions using the static greedy order with the scoring function $\frac{h}{stolen}$, that measures how well a heuristic balances the two objectives of having a high heuristic estimate and stealing low costs from other heuristics (Seipp, Keller, and Helmert 2020). We consider only the single order that is optimized for a high heuristic estimate of the initial state. We optimize the order independent of the transition saturator for a fair comparison. All benchmarks, code, and experimental data have been published online[3].

We write $saturate_1, saturate_2$ for the composition $saturate_{12}$ of saturators where the output of $saturate_1$ is applied to the input of $saturate_2$.

We write $saturate_1 + saturate_2$ to denote that $saturate_2$ is an additional run of cost saturation on the remaining cost after computing a saturated transitioncost partitioning with $saturate_1$. The saturated transition cost functions of both runs form a cost partitioning. Hence, summing up the heuristic estimates of both runs yields an admissible heuristic. Postponing additional runs of cost saturation makes sense if there are potentially unused costs available.

If not explicitly mentioned, a transition saturator composition always uses the transition saturator $spd_t$ first. For example $all_t$ corresponds to $spd_t, all_t$.

**Comparison of transition saturators**  Table 1a shows the pairwise comparison of a set of transition saturators that we obtain using Theorems 2, 3, 4, and the knowledge from previous work about operator saturators. We simulate the saturated transition cost partitioning with the transition saturator $all_t$ (without $spd_t$) and subset-saturated transition cost partitioning and obtain coverage of 974 tasks. Composition with $spd_t$ clearly pays off because $all_t$ has a coverage of 984 tasks and guarantees computing the same heuristics. Selecting the saturated transition cost function that replaces negative infinities by zero $all_t^r$ also pays off because the coverage increases to 1003 tasks, and the estimate of the initial state is worse in only 4 tasks. Preserving the heuristic estimates of reachable state states $reach_t^r$ shows small improvements. Preserving the heuristic estimates of only the initial state $lp_t^r + reach_t^r$ wins the most pairwise comparison for the heuristic estimate of the initial state but solves only 766 tasks. Our best transition saturator is $reach_t^+, perim_t^r + reach_t^r$ that solves 1016 tasks and wins the second most pairwise comparisons for the heuristic estimate of the initial state. The transition saturator $perim_t$ most effectively improves heuristic accuracy.

**Comparison with operator saturators**  Table 1b shows the pairwise comparison of a set of operator saturators

---

[3]https://doi.org/10.5281/zenodo.4065414

| | all$_t$ (without spd$_t$) | all$_t$ | all$_t^r$ | reach$_t^r$ | lp$_t^r$+reach$_t^r$ | perim$_t^r$+all$_t^r$ | reach$_t^+$, perim$_t^r$+reach$_t^r$ |
|---|---|---|---|---|---|---|---|
| all$_t$ (without spd$_t$) | - | **0** | 4 | 17 | 36 | 31 | 33 |
| all$_t$ | **0** | - | 4 | 17 | 36 | 32 | 33 |
| all$_t^r$ | 0 | 0 | - | 13 | 34 | 33 | 34 |
| reach$_t^r$ | **49** | **49** | **69** | - | 30 | 42 | 35 |
| lp$_t^r$+reach$_t^r$ | **370** | **370** | **400** | **397** | - | **137** | **131** |
| perim$_t^r$+all$_t^r$ | **444** | **447** | **475** | **468** | 68 | - | 8 |
| reach$_t^+$, perim$_t^r$+reach$_t^r$ | **445** | **448** | **477** | **474** | 67 | **20** | - |
| Coverage | 974 | 984 | 1003 | 1003 | 766 | 1014 | **1016** |

(a) Comparison of transition saturators

| | all$_o$ | perim$_o$+all$_o$ | all$_t$ (without spd$_t$) | reach$_t^+$, perim$_t^r$+reach$_t^r$ |
|---|---|---|---|---|
| all$_o$ | - | 53 | 153 | 51 |
| perim$_o$+all$_o$ | **485** | - | **370** | 49 |
| all$_t$ (without spd$_t$) | **330** | 219 | - | 33 |
| reach$_t^+$, perim$_t^r$+reach$_t^r$ | **645** | **384** | **445** | - |
| Coverage | 1035 | **1042** | 974 | 1016 |

(b) Comparison with operator saturators

Table 1: Per-task comparison of the initial $h$-value for a subset of all saturator compositions. In every pairwise comparison we consider the tasks for which both transition saturators finished computing the initial heuristic estimate. The entry in row $r$ and column $c$ indicates the number of tasks where $r$ returns a transition cost partitioning with a better initial heuristic value than $c$. Boldface is used to indicate the winner in the pairwise comparison $(r, c)$ and $(c, r)$.

| Coverage | all$_o$ | perim$_o$+all$_o$ | all$_t$ (without spd$_t$) | reach$_t^+$, perim$_t^r$+reach$_t^r$ |
|---|---|---|---|---|
| driverlog (20) | **15** | **15** | 13 | 14 |
| elevators-opt08-strips (30) | 20 | 20 | **21** | 18 |
| elevators-opt11-strips (20) | 17 | 17 | **18** | 15 |
| floortile-opt11-strips (20) | **4** | 3 | 0 | 1 |
| freecell (80) | **65** | **65** | 47 | 47 |
| ged-opt14-strips (20) | 15 | 15 | **19** | **19** |
| miconic (150) | 110 | 110 | 109 | **113** |
| mprime (35) | **28** | 27 | **28** | 27 |
| nomystery-opt11-strips (20) | **20** | **20** | 13 | 13 |
| parcprinter-08-strips (30) | 17 | 18 | 16 | **19** |
| parcprinter-opt11-strips (20) | 13 | 14 | 12 | **15** |
| pipesworld-notankage (50) | 22 | 22 | 22 | **23** |
| scanalyzer-08-strips (30) | **13** | **13** | 10 | 10 |
| scanalyzer-opt11-strips (20) | **10** | **10** | 7 | 7 |
| snake-opt18-strips (20) | **13** | 12 | **13** | **13** |
| tetris-opt14-strips (17) | **11** | 10 | **11** | **11** |
| transport-opt08-strips (30) | **14** | **14** | 13 | 13 |
| transport-opt11-strips (20) | **10** | **10** | 9 | 8 |
| transport-opt14-strips (20) | **8** | **8** | **8** | 7 |
| woodworking-opt08-strips (30) | 20 | 20 | 19 | **26** |
| woodworking-opt11-strips (20) | 14 | 14 | 13 | **18** |
| zenotravel (20) | **13** | **13** | 7 | 7 |
| **Sum (1827)** | 1035 | **1042** | 974 | 1016 |

Table 2: Per domain coverage. Contains all domains where the best operator saturator $perim_o$+$all_o$ solves more tasks than the best transition saturator $reach_t^+$, $perim_t^r$+$reach_t^r$ or vice versa.

and transition saturators. We simulate saturated operator cost partitioning with the operator saturator $all_o$. We choose the best operator saturator $perim_o$+$all_o$ for subset-saturated operator cost partitioning. We simulate saturated transition cost partitioning with the transition saturator $all_t$ (without $spd_t$). We choose the best transition saturator $reach_t^+$, $perim_t^r$+$reach_t^r$ for subset-saturated transition cost partitioning.

Operator saturators solve more tasks compared to transition saturators with 1042 against 1016 solved tasks. This shows that selecting more efficient saturated transition cost functions is crucial for further improvements. The best transition saturator computes significantly fewer heuristics with a lower estimate for the initial state than other saturators (rightmost column). Furthermore, the best transition saturator wins the most pairwise comparisons for the heuristic estimate of the initial state and shows that subset-saturated transition cost partitioning computes more informed heuristics than previous saturators.

**Per domain coverage** Table 2 shows the per domain coverage of a relevant subset of all domains. The best transition saturator $reach_t^+$, $perim_t^r$+$reach_t^r$ performs badly in the domains freecell, zenotravel, and nomystery, and it performs well in domains woodworking, ged and miconic. Furthermore, we see a favor for transition cost partitioning in domains where optimal plans contain an action multiple times. Intuitively, if an optimal plan contains an action multiple times, it applies the action in different states. Otherwise, the plan would not be optimal because it contains a cycle. However, duplicate actions in optimal plans are not necessary for transition cost partitioning to give more informed heuristics.

## Future work

Finding better variable orderings for decision diagrams can further improve the performance and lower the risk of unmanageable large decision diagrams (Keller et al. 2016).

Another important problem to solve is finding mechanisms to select saturated transition cost functions that are computationally easier to handle but still allow us to profit

from more expressive cost assignments. Other selection mechanisms can provide us with polynomial-size guarantees for the representations of transition cost functions during saturated cost partitioning. The question is whether or not such smaller representations are capable of carrying enough context information that allows computing better cost partitioned heuristics.

## Conclusion

We introduced subset-saturated transition cost partitioning that combines saturated transition cost partitioning with the concepts of preserving the heuristic estimates of a subset of states.

Our empirical evaluation shows that more expressive transition cost functions still require too much computational overhead but leads to more informed heuristics. Furthermore, subset-saturated transition cost partitioning lowers the risk of getting heuristics that are worse than heuristics of subset-saturated operator cost partitioning. In other words, the greediness of cost saturation becomes less problematic.

Subset-saturated transition cost partitioning allows selecting from a larger solution set of saturated transition cost functions. Crucial for further improvements is selecting saturated transition cost functions that are computationally easier to handle and still allowing us to obtain better cost partitioned heuristics by considering more expressive cost assignments.

## Acknowledgements

David Speck was supported by the German Research Foundation (DFG) as part of the project EPSDAC (MA 7790/1-1). We sincerely thank the anonymous reviewers for their insightful and detailed comments.

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
