# OpenReview forum: "Subset-Saturated Transition Cost Partitioning for Optimal Classical Planning"
_icaps-conference.org/ICAPS/2020/Workshop/HSDIP — HSDIP 2020_

### Official Review · AnonReviewer2 · 2020-08-14

**Rating:** 6
**Confidence:** 4

**Review:**

The paper merges the existing approaches of subset-saturated cost partitioning
(where only a subset of all states keeps its heuristic value) with saturated
transition cost partitioning (where different transitions with the same
operator label can have different costs). The experimental evaluation shows
that while this approach generally results in better informed heuristics it
does not outperform existing approaches due to the significant computation
overhead caused by the exponential explosion from moving from operator to
transition cost functions.

The topic fits very well in this workshop and I find the approach and its
results interesting, however I would have wished for some more depth especially
in the experimental analysis. It highlights the tradeoff between expansions and
overall time between the best performing subset-saturated *operator* cost
partitioning and subset-saturated *transition* cost partitioning, but a more
general comparison of all tested configurations in terms of coverage is
missing. I would also be interested in how *subset*-saturated transition cost
partitioning compares to saturated transition cost partitioning, since both
share the problem of needing to represent large transition cost functions.

In the theoretical part I'm missing a coherent narrative; there are many
definitions, many of them are similar since they are generalizations of others,
but there is not really a thread tying them all together. I would significantly
shorten the background section (which is over 3 pages long) and focus more on
your generalized definitions, explaining the differences to existing definitions
without writing them down as well. For example, Definition 15 and 17 are very
similar, the major difference being that Definition 17 talks about transitions
rather than operators. I would only include Definition 17 and (shortly) discuss
that this Definition is similar to the one for subset-saturated *operator* cost
partitioning except for ..., and whether or not these changes introduce any
problems we need to be aware of (which I think is never the case?). Similarly,
many definitions in the background itself are redundant, for example Definition
10 (Transition Cost Partitioning) and 12 (Operator cost partitioning).

The description of the transition saturators on the other hand is somewhat
short, especially for the saturator you newly introduce. I would put more focus
on this saturator (since the others are fairly straightforward generalizations
of existing operator saturators) and explain in more detail how it works. For
example, I don't understand what you mean with setting the remaining smallest
cost of 0 to the abstract transition. Does that mean you set the cost to 0? Is
that safe (i.e. won't you introduce shortcuts this way)?

Overall I find the topic interesting but I do think the presentation of the
paper could be improved significantly, the description of the new transition
saturator needs to be improved, and the experimental evaluation should be more
extensively discussed, especially the comparison to saturated transition cost
partitioning.


Questions to the authors:
 1) Could you give a more in-depth explanation of the spd saturator?
 2) You say you always apply spd first. Does this mean that for example all_t
 is actually spd_t,all_t?
 3) Can you elaborate on the overall experimental results, specifically the
 comparison with saturated transition cost partitioning?


Minor comments
 - Cost Partitioning (and all its variations) -> cost partitioning
 - Classical Planning -> classical planning
 - Introduction: "Seipp and Helmert showed that ... (Seipp, Keller and
 Helmert)" Why do you cite two papers here? I don't understand if both papers
 showed the same thing (then why not cite both the same way?), or, if not, what
 the second citation is for.
 - Introduction: You say that unlike previous approaches your approach does not
 leave any costs unused. But you don't back that claim anywhere in the
 theoretical part.
 - Definition 2: "The length of \pi denoted by |\pi|, is n" -> \pi, denoted by
 (missing comma)
 - Definition 2: "\pi is called a goal path, if" --> goal path if (remove comma)
 - Planning tasks: "The set of operators induce a" ... --> induce*s* a
 - Planning tasks: "A finite set of operators (or actions) O where..." -> The
 main sentence is unfinished.
 - Abstractions: "every goal path in the abstract transition system corresponds
 to a goal path in the concrete transition system" -> this should be the other
 way around.
 - Abstractions: I don't understand the purpose of the last paragraph. You use
 A* with admissible heuristics within the abstraction? If not, why is this
 sentence there?
 - Definition 15 + 17: o/tcf_i = saturate_i(...), not saturate_{i-1}
 - The description of how the heuristic order is optimized is very unhelpful,
 it is not explained what h/stolen means and I don't understand how it can be
 optimized independent of the used transition saturator
 - Page 8: try to avoid a paragraph break after a single word
 - References: What is "2014. IPC-8 planner abstracts."? Is this used anywhere?
 - References: Keller et al. 2016a and Keller et al. 2016b are the same paper.
 - References: Kissmann and Edelkamp 2014: IPC in title not capitalized

---

> ### Author Response · Authors · 2020-08-24
> **Answers to questions**
>
> Dear AnonReviewer2,
>
> Thank you for your review!
>
> > In the theoretical part I'm missing a coherent narrative; there are many definitions, many of them are similar since they are generalizations of others, but there is not really a thread tying them all together. I would significantly shorten the background section (which is over 3 pages long) and focus more on your generalized definitions, explaining the differences to existing definitions without writing them down as well. For example, Definition 15 and 17 are very similar, the major difference being that Definition 17 talks about transitions rather than operators. I would only include Definition 17 and (shortly) discuss that this Definition is similar to the one for subset-saturated operator cost partitioning except for ..., and whether or not these changes introduce any problems we need to be aware of (which I think is never the case?). Similarly, many definitions in the background itself are redundant, for example Definition 10 (Transition Cost Partitioning) and 12 (Operator cost partitioning).
>
> Thank you for these precise suggestions. We will make the background part more compact.
> You are right that it is perfectly fine to use the definition of transition saturator with a sentence that allows us to define an operator saturator.
>
> > 1. Could you give a more in-depth explanation of the spd saturator?
>
> The computation of the abstraction heuristic from a given abstraction and remaining transition cost function requires the computation of the abstract transition cost function.
> The transition saturator spd_t computes the abstract transition cost function tcf' during the goal distance computation with Dijkstra's algorithm. Initially, all abstract transition costs are unknown.
> When Dijkstra's algorithm expands an abstract transition t = <s,l,s'> and it holds that dist[s] <= dist[s'] then the abstract transition cost tcf'[t] = 0 does not introduce shortcurts because dist[s] <= disŧ[s'] + tcf'[t].
> Otherwise, query a request to the data structure to obtain tcf'[t].
>
> > 2. You say you always apply spd first. Does this mean that for example all_t is actually spd_t,all_t?
>
> Yes. In the composition with each transition saturator except for lp_t, the composition with spd_t results in the exact same heuristic. The reason for a different result in the composition with lp_t is that it gives lp_t fewer possibilities to adapt the heuristic estimates and the saturated transition cost function.
>
> > 3. Can you elaborate on the overall experimental results, specifically the comparison with saturated transition cost partitioning?
>
> We will improve the comparison against saturated transition cost partitioning. We simulate saturated transition cost partitioning with subset-saturated transition cost partitioning and the saturator all_t (without spd).
> We will add the saturator all_t (without spd) to the comparison. It has a coverage of 974 tasks.
> The saturator all_t solves 984 tasks.
> With additional relaxation, all_t' solves 1002 tasks with nearly the exact same heuristics as all_t (Table 1).
> The saturator perim_t,all_t'+all_t' solves 1012 tasks.
> We will exchange perim_t,all_t'+all_t' with perim_t,reach_t'+reach_t' that solves 1016 tasks with slightly better heuristics.
> The operator saturator perim_o,all_o+all_o solves 1042 tasks.

---

### Official Review · AnonReviewer1 · 2020-08-17
**Good paper, topic fits workshop, but experimental section could be improved**

**Rating:** 6
**Confidence:** 4

**Review:**

The submission presents an extension of the subset-saturated cost partitioning framework for optimal classical planning. This extension considers transition- instead of operator- cost functions, very much in the same way as (Keller et al. 2016a) does for the standard saturated cost partitioning variant. The authors present an empirical evaluation of the technique that shows that while the obtained heuristics are more informed than their *operator* counterpart, overall the technique does not pay off.

The submission clearly fits the scope of the workshop, and while it reads a bit incremental, I think it can make up for an interesting presentation and discussion. I also think the background and definition sections are too lengthy, especially when compared to the experiments section, and taking into account that many definitions are straight-forward adaptations of their operator equivalents. I would suggest the authors could focus more on highlighting the differences with the latter. I definitely missed a bit more detail and analysis in the experiments section (which are by the way the benchmarks being used? ). Besides the overall conclusion on wheter the technique pays off overall or not, it would be great if the experiments could teach us something and illustrate in what kind of domains it works better/worse, and why that is the case.

---

> ### Author Response · Authors · 2020-08-24
> **Answers to questions**
>
> Dear AnonReviewer1,
>
> Thank you for your review!
>
> > The submission clearly fits the scope of the workshop, and while it reads a bit incremental, I think it can make up for an interesting presentation and discussion. I also think the background and definition sections are too lengthy, especially when compared to the experiments section, and taking into account that many definitions are straight-forward adaptations of their operator equivalents. I would suggest the authors could focus more on highlighting the differences with the latter. I definitely missed a bit more detail and analysis in the experiments section (which are by the way the benchmarks being used? ). Besides the overall conclusion on wheter the technique pays off overall or not, it would be great if the experiments could teach us something and illustrate in what kind of domains it works better/worse, and why that is the case.
>
> Thank you for the suggestions on how to make the background section more compact. We will incorporate your suggestions.
> The benchmark set consists of all 1827 planning tasks from the optimization track of the 1998-2018 IPCs that do not have conditional effects.
>
> We will provide information about specific domains where the techniques show significant differences. Example domains where we see the most improvements are elevator, woodworking, and ged.
>
> Transition cost partitioning helps especially in cases where optimal plans apply actions multiple times:
> An optimal plan that applies an action multiple times does this in different application contexts. Otherwise, the optimal plan would cycle between states and would not be optimal. If different abstractions focus on these contexts, then transition cost partitioning can help.
>
> In a comparison of operator saturator perim_o,all_o+all_o with transition saturator perim_t,reach_t'+reach_t', we see that in 64% of all tasks where either one of both saturators returned a plan that contains duplicate actions, the expansions until the last f-layer is smaller for the transition saturator. In the same comparison, we see that in 41% of all tasks where both saturators returned a plan that contains no duplicate actions, the expansions until the last f-layer is smaller for the transition saturator.
> We take this as an indicator that saturated transition cost partitioning (or transition cost partitioning in general) can help in tasks that require the application of the same action multiple times.

---

### Comment · Program_Chairs · 2020-09-14
**Final Decision: Accept**

Dear Authors,

Thank you very much for your submission. We are happy to inform you that we have decided to accept it and we look forward to your talk in the workshop. You will receive additional information per mail in the coming days.

Best,
The HSDIP'20 team

---

### Decision · Program_Chairs · 2020-09-30

Accept